# LUNARFM: A MULTIMODAL REPRESENTATION OF THE MOON'S SURFACE

**Jakob Gawlikowski**[*†]
German Aerospace Center (DLR)
Remote Sensing Technology Institute
Wessling, Germany

**Marc Girona-Mata**[*†]
University of Cambridge
Department of Engineering
Cambridge, United Kingdom

**Sumit Goski**[†]
SPAIDERS SPACE
Esch-sur-Alzette,
Luxembourg

**Gautier Bardi de Fourtou**[†]
Mines Paris - PSL University
Paris, France

**Abigail Calzada-Diaz**
European Space Resources Innovation Center (ESRIC)
Sanem, Luxembourg

**Sylvester Kaczmarek**[†]
Imperial College London
London, United Kingdom

**Raúl Ramos-Pollán**[†‡]
Universidad de Antioquia
Faculty of Engineering
Medellín, Colombia
raul.ramos@udea.edu.co

## ABSTRACT

The renaissance of lunar exploration is largely driven by the dual goal of assessing the Moon's resource potential and establishing a sustained human presence. However, current resource mapping efforts remain fragmented due to heterogeneous, multi-instrument datasets that expose fundamental limitations in existing analysis pipelines. Here, we introduce LunarFM, a first-generation foundation model that takes a step toward unifying heterogeneous lunar remote-sensing observations from multiple instruments in a shared representation space, integrating data from six instruments across three orbital missions with a total of 18 input channels. It is trained via multimodal self-supervised pre-training. The LunarFM framework consists of (1) a machine-learning-ready dataset of co-registered multimodal chips spanning $0.5° \times 0.5°$ of lunar latitude and longitude, covering $\pm 70°$ latitude; (2) a pre-trained multimodal masked autoencoder trained on these inputs, and a companion embedding dataset providing a joint 768-dimensional representation of lunar surface properties; and (3) three illustrative downstream applications, including similarity search and few-shot resource mapping (e.g., ilmenite) to demonstrate the framework's potential. LunarFM is intended to lower the barrier to entry for lunar scientific investigation and resource-oriented analysis, with the expectation that future work will extend and rigorously benchmark the framework. All code and data is available at https://lunarfm.spaceml.org

## 1 INTRODUCTION

Over the last two decades, the number of missions to the Moon has skyrocketed. Besides scientific interests, this renewed activity is primarily driven by two goals: exploring the Moon as a potential source of valuable resources and establishing a long-term, sustained lunar station as a base for deeper space exploration. In 2009, the Indian mission Chandrayaan-1 detected signatures of $H_2O$ and $OH$ in the southern circumpolar regolith (Pieters et al., 2009; Colaprete et al., 2010); this was confirmed a little later by NASA's Lunar Crater Observation and Sensing Satellite (LCROSS) mission, which unambiguously detected water ice in Cabeus Crater close to the lunar south pole (Neish et al., 2011).

---

[*]equal contribution
[†]Work done within the FDL research sprint.
[‡]Corresponding Author

Besides water ice, the Moon is known to contain many other useful materials. Ilmenite (titanium dioxide, $TiO_2$), silicon, and aluminium (e.g,. $Al_2O_3$) are abundant in lunar regolith and can be used for construction (e.g., solar panels; Cuervo-Ortiz et al., 2025) and life-support applications (e.g., oxygen, rocket fuel Sargeant et al., 2020). Other elements, such as rare-earth metals (Hedrick, 2023) and helium-3, could be used in advanced technologies and energy production. We note, however, that the current LunarFM framework is limited to latitudes between $\pm 70°$, thereby excluding the polar regions.

Lunar geology and resource maps are typically derived from multiple orbital missions that collect heterogeneous datasets, including spectral, gamma-ray, neutron, and topographic measurements, amongst others (e.g., Sato et al., 2017). From these primary observations, a range of secondary and higher-level products is generated through mission-specific processing pipelines. In practice, however, most existing lunar maps and resource assessment studies rely on a limited combination of datasets, rather than fully exploiting the large amount of available information. This is not due to a lack of available data, but rather to structural challenges in data integration. Datasets differ substantially in spatial and temporal resolution, observation geometry, sensitivity, radiometric and signal-to-noise characteristics, processing assumptions, and also in the spatial-temporal coverage of the lunar surface. As a result, combining these datasets in a physically-consistent and statistically-rigorous manner is not trivial and requires methodological choices that are specific to each dataset combination.

To this end, we introduce LunarFM, a first-generation multimodal representation framework for the Moon. To our knowledge, LunarFM is the first attempt to leverage data from multiple lunar instruments within a shared representation space, with the goal of supporting a suite of scientifically relevant downstream tasks. LunarFM combines six data modalities, each corresponding to a different instrument and acquired via three different lunar remote-sensing missions, yielding a total of 18 input channels/bands. We deliberately scope LunarFM as a proof-of-concept: our primary aims are to establish the data infrastructure, propose a training protocol, and identify a suite of evaluation tasks for future work to build upon.

Our contribution is a data stack that consists of

1. a machine learning-ready dataset (LunarChips) that includes data from three lunar orbital remote-sensing missions and six instruments/modalities for LunarFM pre-training, along with additional datasets used for evaluation, divided into chips that extend 0.5 degrees of latitude and longitude each;

2. a pre-trained multimodal masked autoencoder and companion embedding dataset (LunarEmbeddings), providing 768-dimensional dense representations for each (18-channel) 0.5° input chip; and

3. an initial set of evaluation experiments and illustrative downstream applications, including multimodal input reconstruction, unsupervised exploration of embedding representations, and example use cases that demonstrate the potential of the learned representations.

In addition to the implementations, models, and datasets, we provide tutorials for a straightforward introduction to the hands-on application of these resources.

In the following, we introduce the data sources included in the provided dataset and used for training LunarFM, we describe the model's multimodal architecture and training strategy, we explore the resulting embedding representations, and we present illustrative application examples including similarity search and resource mapping for minerals as ilmenite ($TiO_2$), magnesium oxide ($MgO$), or iron (II) oxide ($FeO$), accompanied by notebooks for hands-on exploration. We discuss design choices and current limitations throughout, and frame open questions for future work in Section 5.

## 2 BACKGROUND

Foundation models (FMs) are large-scale, pre-trained models that have demonstrated remarkable capabilities across diverse domains (Wang et al., 2022; Tao et al., 2023). Their rise has also been seen in Earth observation (EO), with recent developments showing promising results (Jakubik et al., 2025; Astruc et al., 2025; Tseng et al., 2025). Building on these advances, geospatial foundation models (GFMs) such as CROMA (Fuller et al., 2023), DOFA (Xiong et al., 2025), TerraMind (Jakubik

Table 1: Overview of selected orbital remote-sensing instruments and data modalities integrated into the LunarFM model. Note that Diviner Rock Abundance (RA) is treated as a separate input group during training, yielding seven input groups in total.

| Instrument | Mission | Modality | Channels | Description |
|---|---|---|---|---|
| LROC WAC | LRO | Multispectral imagery | 7 | Photometrically Normalised Mosaic via Hapke Parameter Maps |
| LOLA | LRO | Topography elevation | 1 | Surface digital elevation model (DEM) |
| DIVINER | LRO | Thermal emission | 3 | Regolith temperature, rock abundance, bolometric temperature |
| Mini-RF | LRO | Radar | 2 | Circular polarization ratio and S1 (first Stokes parameter) |
| Ka-band Lunar Gravity Ranging System | GRAIL | Gravitational anomaly | 4 | Free-air gravity anomaly, Bouguer anomaly, disturbance, and measurement uncertainty |
| UVVIS Camera | Clementine | Albedo | 1 | Brightness measured at the 750 nanometer (nm) wavelength |

et al., 2025), AnySat (Astruc et al., 2025), and Galileo (Tseng et al., 2025) have emerged, integrating multiple data sources such as optical, radar, digital elevation models (DEMs), land-cover, and text, into unified architectures capable of sensor-agnostic understanding across diverse downstream tasks.

FMs mainly provide two key advantages relevant to planetary science: (1) self-supervised pre-training captures general features of the underlying data, including correlations among modalities and spatial regions, without requiring labelled data; and (2) the resulting *embeddings* enable unsupervised analysis and data-efficient training of downstream tasks, both critical properties in a domain where labelled data is scarce.

To the best of our knowledge, no prior work has attempted multimodal representation learning across this breadth of lunar instruments. Sander et al. (2025) trained a transformer model on grayscale images, surface normals, DEMs, and albedo maps, with a focus on reconstructing DEMs and albedo from reflectance imagery. LunarFM extends this direction by integrating a broader set of physically distinct modalities, spanning optical, thermal, radar, topographic, and gravitational data, within a shared representation space designed to support a range of downstream tasks.

## 3 TECHNICAL SETUP

### 3.1 DATASET PREPARATION

For training LunarFM, we combine data from six modalities across three missions: the Lunar Reconnaissance Orbiter (LRO), the Gravity Recovery and Interior Laboratory (GRAIL), and the Clementine missions. An overview of the considered missions and sensing instruments is provided in Table 1, and selected examples are shown in Fig. 1. Further details are included in Appendix A.

Data was obtained from NASA's Planetary Data System (PDS) Geoscience Node [1], either from global mosaics or other derived products as available at PDS. We extract $0.5° \times 0.5°$ longitude-latitude chips from the data using a spherical grid discretisation of the Moon's surface and store them as individual GeoTIFF files. For training and inference, each channel is normalised independently using band-wise mean and standard deviation computed across the training set. Due to data availability, resolution, and distinct solar illumination effects, we focus on the region between -70° and +70° latitude, resulting in a total of 201,600 chips. We further apply a band-wise split to separate grid cells into training, validation, and test sets of 117,024, 38,715, and 45,861 samples, respectively (see Appendix A.3). The band-wise split is used to mitigate spatial leakage, thereby enabling robust

---

[1]https://pds-geosciences.wustl.edu/dataserv/moon.html

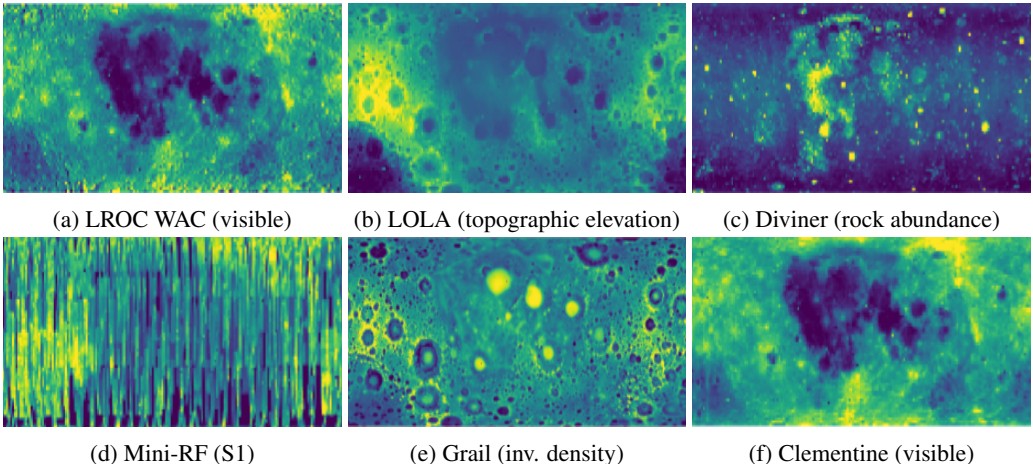

(a) LROC WAC (visible)          (b) LOLA (topographic elevation)          (c) Diviner (rock abundance)

(d) Mini-RF (S1)          (e) Grail (inv. density)          (f) Clementine (visible)

Figure 1: Maps showing single-channel examples for the six instruments included in the LunarChips dataset and used for the pre-training LunarFM, covering most of the Moon's surface, from -70° to +70° latitude. For LunarFM, Diviner Rock Abundance is treated as an individual seventh input.

evaluation of the model's generalisability and reducing overfitting. The resulting data is part of our LunarChips dataset.

Additionally, LunarChips includes products for downstream applications. These include global elemental and compositional maps derived from gamma-ray spectroscopy (Prettyman et al., 2006), WAC-derived ilmenite estimates, and impact crater annotations used for crater-focused analyses. We further incorporate the delineations of the high-ilmenite ($TiO_2$) region proposed by Diaz & Keszthelyi (2025), which we treat as an expert-curated reference set for low-label resource mapping experiments. Collectively, these layers enable consistent, chip-aligned supervision and evaluation across tasks (e.g., mineral regression, few-shot resource screening, and crater-related tasks) while keeping the representation-learning stage strictly self-supervised for the six primary sensing modalities.

## 3.2 Model Architecture and Training

To compute the joint representation, we use a multimodal multi-masked autoencoder (MultiMAE; Bachmann et al., 2022) with visual transformer backbones as the encoder and decoder. The input chips are grouped into seven groups (with each modality grouping channels corresponding to the same instrument, except for Diviner, which is split into two modalities) and divided into patches of $8 \times 8$ pixels, followed by a modality-wise linear projection of each patch to an input token with dimension 768. Linear projections are optimised during training individually for each input data source. Spatial information is added using sinusoidal positional embeddings, which encode the patch's relative position within the chip. Data source information is preserved by modality-wise linear mappings that learn to encode modality-specific bias values. In addition to the patch-level tokens in the input data, we extend the input representation with a global token that is independent of modality or patch location. In the embedding space, this global token serves as the chip-level embedding. Following Bachmann et al. (2022), the input tokens are processed by a joint encoder network based on a visual transformer, which yields latent embedding vectors that capture both single-modality and cross-modality information. Given the available embeddings and the global (modality-independent) token, lightweight modality-wise decoders aim to reconstruct the original input. The final model consists of 109.8 million parameters.

During training, we augment the training data by first slicing the Moon into 2° chips and then randomly selecting 0.5° footprints within each 2° chip. Missing values are set to zero and excluded from the reconstruction loss, such that NaN regions do not contribute to the gradient during training. Chips are aligned and resized modality-wise to $112 \times 112$ pixels. The patch size is set to $8 \times 8$ pixels, yielding 196 patches ($14 \times 14$) per 0.5° input chip. During training, we randomly mask 85% of the input patches across all input modalities, while the decoder reconstructs the full input

domain, including masked locations, thereby favouring the learning of intra- and cross-modality correlations. Training minimises the mean squared reconstruction error, weighted equally across modalities, using the Adam optimiser with an initial learning rate of $10^{-4}$, reduced every 50,000 iterations. The model is trained for approximately 500,000 iterations with a batch size of 32 per GPU on three NVIDIA H100 GPUs.

## 4 EVALUATION

### 4.1 RECONSTRUCTION ERROR

First, we evaluate the reconstruction performance as a proxy for the quality of the learned representations, noting that perfect reconstruction is not the model's primary goal. Figure 2 shows the average squared error in the reconstruction for the individual modalities, both when the modality is present and when it is withheld. As expected, the error is higher when reconstructing a missing modality. For Mini-RF, the error is higher than for the others, which can be explained by the nature of Mini-RF and various NaN values in the data (see also Figure 1 (d)). Figure 3 shows an example for a reconstruction with unmasked and 80% masked-out inputs, underlining the learned correlations among the modalities.

### Reconstruction Error (mean ± std)

| | Diviner (RA) | Diviner (Temp) | LRO LOLA (DEM) | Grail | Mini-RF | LRO WAC | Clementine |
|---|---|---|---|---|---|---|---|
| With modality | 0.092 ±0.085 | 0.008 ±0.004 | 0.001 ±0.001 | 0.002 ±0.003 | 0.420 ±0.190 | 0.026 ±0.015 | 0.022 ±0.015 |
| Without modality | 0.203 ±0.171 | 0.055 ±0.035 | 0.007 ±0.008 | 0.266 ±0.304 | 0.440 ±0.210 | 0.059 ±0.050 | 0.067 ±0.069 |

Figure 2: Modality-wise reconstruction errors for reconstructions based on all input modalities (top row) and with the reconstructed modality not in the input (bottom row). The reconstruction error is expressed as mean squared error with respect to standardised inputs with zero mean and unit variance, based on the training data statistics. NaN input pixels are excluded from the computation.

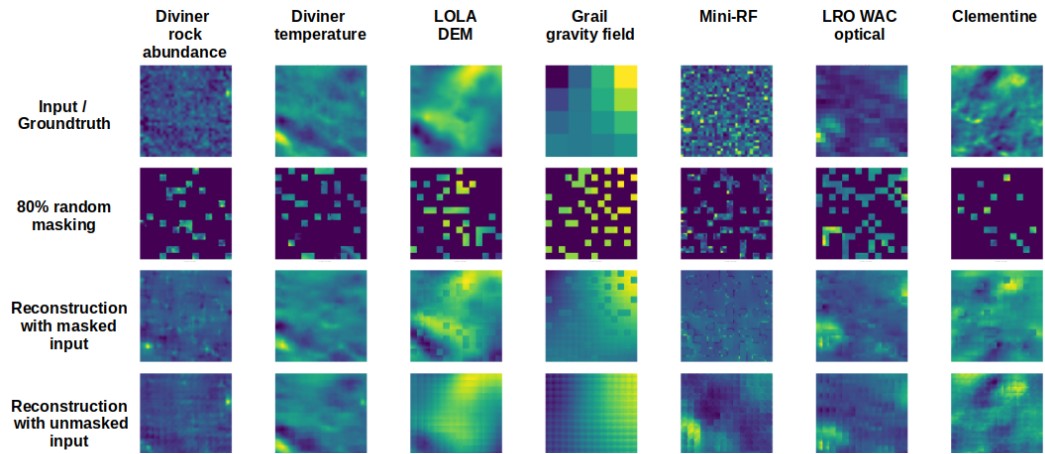

Figure 3: An example of modality-wise reconstruction of a 112 by 112 pixel chip with 80% masking (third row) and without masking (fourth row). Note that the model is trained with 85% masking.

### 4.2 LUNAR EMBEDDINGS

**Correlation.** To assess first insights into the information density of the 768-dimensional feature space, we computed the Pearson correlation coefficient ($r$) between all pairs of embedding dimen-

sions. Figure 4 (left) displays the distribution of these coefficients, which decays rapidly, with only 2.8% of embedding component pairs exhibiting an absolute correlation $|r| \geq 0.5$. This low degree of cross-correlation indicates that the learned embedding dimensions are largely orthogonal, suggesting the model avoids trivial linear redundancy, however we note that low pairwise (linear) correlation is a necessary rather than sufficient condition for information richness.

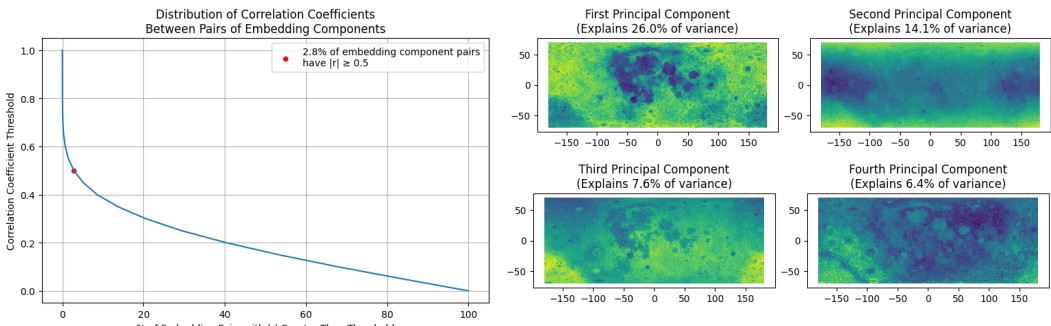

Figure 4: Analysis of the internal structure of the embeddings. **Left:** the distribution of correlation coefficients, highlighting a rapid decay and low redundancy between dimensions. **Right:** Maps of the first four principal components (PCs), which cumulatively explain 54.1% of the variance.

**Dimensionality reduction.** We use two distinct dimensionality reduction techniques, namely Principal Component Analysis (PCA) and Uniform Manifold Approximation and Projection (UMAP), to visualise the high-dimensional embedding space with the intent to gain an intuitive understanding of how the embeddings are able to capture geological features on the Moon. We employ this combination to capture different aspects of the data topology: PCA reveals global linear variations, whereas UMAP exploits non-linearity to highlight both local structure whilst retaining a global continuous manifold. We extracted the first four principal components (PCs) to analyse the dominant linear variations in the data. Together, these four components account for approximately 54.1% of the total variance in the dataset and are visualised in Figure 4 (right). The first component alone accounts for 26.0% of the variance, with subsequent components (PC2 through PC4) explaining 14.1%, 7.6%, and 6.4%, respectively. The spatial distribution (and coherence) of these components suggests that the orthogonal directions of maximum variance in the embedding space correspond to large-scale lunar surface features. These four components appear to be influenced by distinct input data, namely surface albedo and elevation (PC1), effective albedo and surface temperature (PC2), gravity and roughness (PC3), and a strong gravity component (PC4).

To investigate non-linear relationships, we project the embeddings into two dimensions using UMAP. The two components visualised in Figure 5 (left), show a continuous manifold structure. The spatial distribution of the two UMAP components aligns with surface features, indicating that the embeddings encode smooth transitions among lunar terrains while maintaining distinct boundaries between disparate geological zones. Figure 5 also shows the colouring of the space by ilmenite ($TiO_2$) concentration, revealing a small, relevant subregion in this distribution. These structures also show differentiated grouping when using other reference sets for different minerals. When these two UMAP dimensions are mapped back onto the lunar surface (right part of Figure 5), they exhibit strong spatial contiguousness, differentiating major geological regions despite the model having no explicit coordinate input during inference. Overall, this provides evidence that the embeddings are sensitive to different physical properties.

**Clustering.** Finally, we apply k-means clustering ($k = 5$) directly to the 768-dimensional embeddings. Figure 6 illustrates the resulting cluster map and the distribution of chips across clusters. The resulting clusters map to spatially-coherent (and geographically meaningful) regions without any neighbourhood constraints enforced during the clustering process. The distribution of chips shows that while some clusters represent vast, dominant terrain types (e.g., cluster 2), others capture rarer surface characteristics (e.g., cluster 3), suggesting that the model can discriminate between common and unique lunar features.

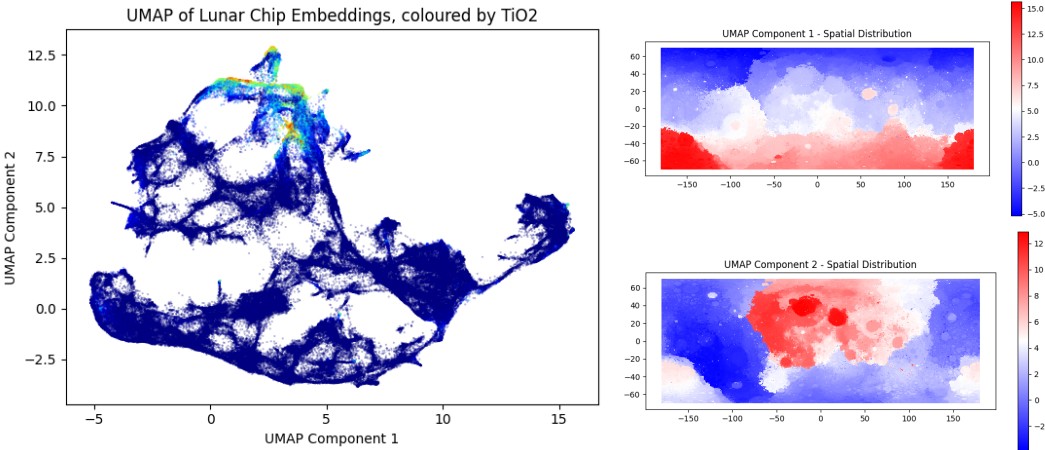

Figure 5: UMAP analysis of lunar embeddings. **Left:** 2D scatter plot of the projected embeddings, coloured by WAC-derived ilmenite ($TiO_2$) percentage. Observe how chips with high $TiO_2$ are naturally separated in this projection, whilst nowhere in the training process $TiO_2$ information was shown to the model. **Right:** Spatial distribution of the first and second UMAP components mapped to the lunar grid. Observe how major geological features of the moon naturally arise.

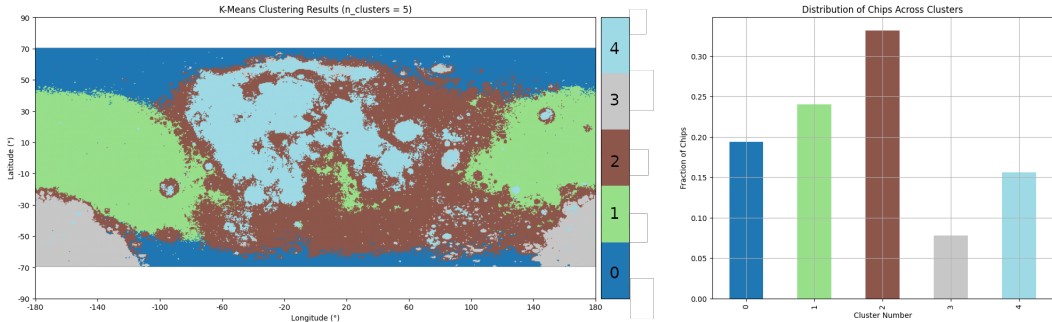

Figure 6: Unsupervised K-Means clustering ($k = 5$) of the lunar embeddings, revealing five distinct clusters: 0) circumpolar regions, 1) equatorial farside, 2) nearside highlands, 3) South Pole-Aitken Basin, and 4) mare regions. Left: The spatial distribution of the assigned clusters. Right: The fraction of total lunar chips assigned to each cluster.

### 4.3 SIMILARITY SEARCH

Latent spaces of large unsupervised trained models have been shown to provide good premises for an efficient similarity search. If images share information in their inputs, then this information should also lead to a corresponding similarity in the embeddings that encode these features. In Figure 7 an example for similarity search based on the L2 distance among the z-score normalised embeddings is shown. For the moment, we keep this assessment qualitative as shown in the figure, whilst a more formal retrieval metric on the Moon might be developed in the future.

### 4.4 EXAMPLE DOWNSTREAM APPLICATIONS

In this section, we present three illustrative downstream applications that demonstrate how LunarFM embeddings can serve as a starting point for resource-oriented analysis. The experimental setting is as follows. First, we obtain embeddings for all lunar chips. Then, we randomly select a subset of 20,000 chips from the training split. Subsequently, we train a simple regression model (random forest) using the embeddings as inputs to predict the average mineral content per chip on the training data subset. Finally, we use the trained random forest model to predict mineral content for the remaining chips. Figure 8 shows the different mineral compositions we used to test this ap-

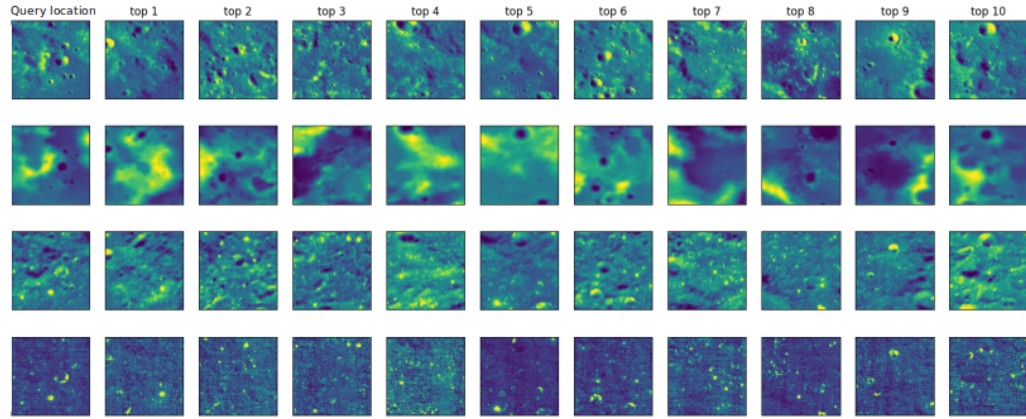

Figure 7: An example of a similarity, four modalities of the query location are shown together with the top ten matches. The rows from top to bottom show LRO WAC, LOLA DEM, Diviner Regolith Temperature, and Diviner Rock Abundance.

proach. Again, our intent is to show that a simple regression model on our embeddings can produce meaningful mineral distribution maps. Future work could be targeted to specific applications and benchmarks (prospecting, feature characterisation, etc.)

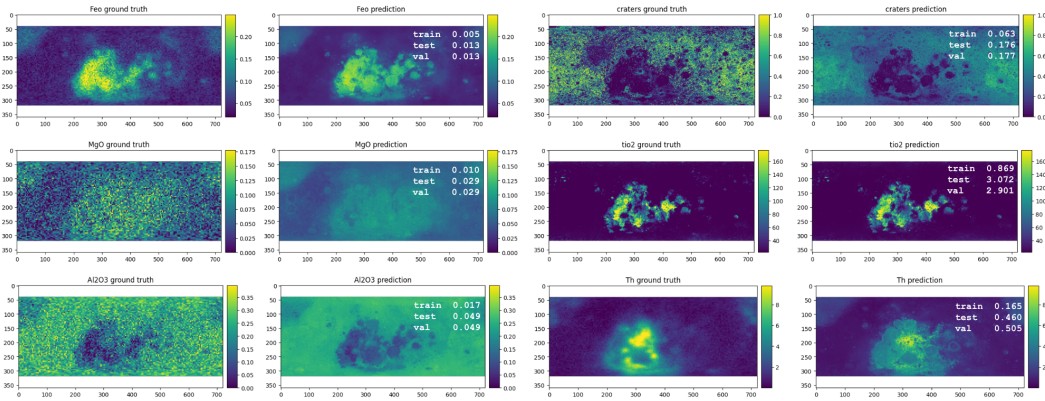

Figure 8: Results on different downstream tasks predicting the presence of different minerals on the Moon's surface, based on random forest. In the upper right of the predicted maps, the Mean Absolute Errors of the predictions are stated. Note that latitude axis is inverted.

Furthermore, we evaluate LunarFM in a more realistic scenario, in which we have access to only a handful of expert-curated data points to train a downstream regression model. To this end, we use the recent work by Diaz & Keszthelyi (2025), in which the authors developed a descriptive/conceptual model to map areas on the Moon with high concentrations of ilmenite ($TiO_2$). These high-titanium areas intersect with approximately 800 half-degree lunar chips. From these, we randomly select 5 chips as positive examples and another 5 from the lower percentile of $TiO_2$ concentration as negative examples. In this way, we have approximated the scenario in which an expert selects 10 chips with known (very high and very low) mineral concentrations. We then train a random forest that takes LunarFM chip embeddings as inputs and predicts $TiO_2$ concentration across the lunar surface. We repeat the experiment 20 times and report the mean and standard deviation of the correlation coefficient between the ground truth and the predictions. In parallel, we perform a similar experiment, but instead of using an expert-curated (and thus highly informative) training dataset, we randomly select 10 chips from the Moon's surface. Fig 9 shows the results. However, when combined with expert advice on selecting high-quality training locations, the downstream embedding training yields substantially better predictions, with 10× lower variability.

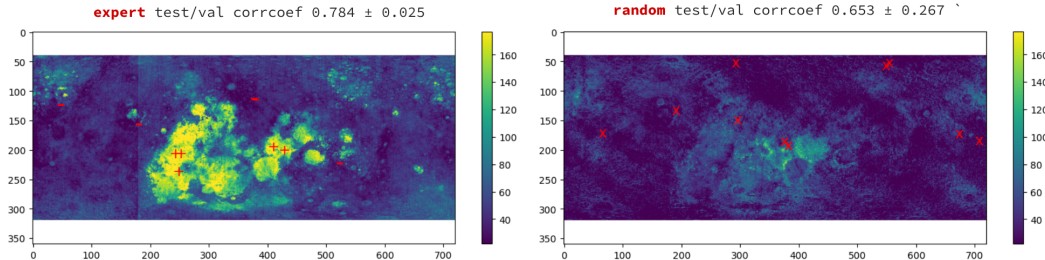

Figure 9: Random forest models trained with expert location selection vs. random selection. The red plus and minus signs on the left-hand chart indicate the expert's positive and negative choices in one of the expert runs. Red crosses on the right map show the random choices on one of the random runs. You can compare this with the $TiO_2$ ground truth map in Fig. 8. The latitude axis is inverted.

## 5 DISCUSSION, LIMITATIONS & CONCLUSION

The evaluations presented above provide encouraging, if preliminary, evidence that multimodal self-supervised pretraining produces representations with meaningful structure for lunar science applications. The reconstruction results indicate that the model learns both intra-modality and cross-modality correlations, and even when a full modality is withheld at inference, the model produces plausible reconstructions by drawing on complementary information from the remaining inputs. However, reconstruction quality varies substantially across modalities and examples. The Mini-RF error likely reflects the presence of NaN pixels in the training data and noise in radar backscatter. We also note that Diviner Rock Abundance (RA) occasionally exhibits unexpected reconstruction behaviour, with outputs that appear to track Diviner temperature products rather than rock abundance, likely due to the strong correlation between these two inputs. The embedding analyses provide qualitative evidence that the representation space organises along physically meaningful dimensions. PCA components exhibit spatially coherent patterns broadly consistent with known geological terrain types. UMAP projections reveal a continuous manifold with differentiated sub-regions, and colouring by $TiO_2$ concentration identifies a distinct cluster of high-titanium chips within this space. Unsupervised k-means clustering yields spatially contiguous groupings without geographic constraints, suggesting that the embeddings encode large-scale surface variability in a geographically coherent manner. We note that evaluating learned representations in a domain with scarce and spatially irregular ground truth is inherently challenging. Identifying resource-relevant structure in an unsupervised setting proved non-trivial: while high-$TiO_2$ regions form a discernible subregion in embedding space, separating specific resource signatures from broader terrain patterns requires targeted evaluation protocols that go beyond what is established here.

**Downstream Use Cases.** The downstream experiments are presented as illustrative use cases that demonstrate the practical potential of the LunarFM embeddings, rather than as exhaustive benchmarks. The mineral mapping experiment (Section 4.4) yields spatially coherent predictions of mineral concentrations including $FeO$, $TiO_2$, $MgO$, and $Al_2O_3$, with reasonable agreement between predicted and reference maps, illustrating that the embeddings are directly useful for regression-based compositional mapping even with limited supervision. In a more operationally realistic scenario, we simulated the workflow of an expert selecting a small number of reference locations with known high and low $TiO_2$ concentrations. Using only 10 expert-selected chips for training, the random forest achieves a correlation of $0.784 \pm 0.025$ with the ground-truth map, compared to $0.653 \pm 0.267$ when the same number of chips is selected at random. The substantially lower variance in the expert condition highlights an important practical finding: the value of the embeddings is amplified when combined with domain knowledge in sample selection, reflecting a realistic workflow for targeted resource assessment where a geologist contributes a small number of high-confidence annotations and the embedding space generalises across the remainder. Querying the embedding space using the L2 distance retrieves chips that are visually and geophysically similar across multiple modalities. This capability could support prospecting workflows in which an analyst identifies a region of interest and seeks analogous surface conditions elsewhere on the Moon, without requiring explicit spectral or compositional criteria.

**Limitations.** LunarFM is a first-generation framework, and several aspects reflect both the practical constraints under which it was developed and the fundamental challenges of the lunar domain. The dataset is restricted to $\pm 70°$ latitude due to gaps in modality coverage at high latitudes and distinct illumination-related challenges, thereby excluding the lunar south pole, i.e., the primary region of interest for water-ice prospecting operations. The Mini-RF data modality contains substantial NaN regions from incomplete orbital coverage; these NaN regions are masked during training, which limits the model's ability to fully exploit radar backscatter information and, in turn, contributes to the substantially higher reconstruction error observed for this modality. Chip-level embeddings collapse a $0.5°\times0.5°$ surface area into a single vector, corresponding to roughly 15 km in the north-south direction, with the east-west extent varying from $\sim$15 km at the equator to $\sim$5 km near 70° latitude. This varying footprint size implies that embeddings represent different physical extents at different latitudes, complicating direct comparisons across the coverage area. Given the non-inclusion of high-resolution products, such as LROC Narrow Angle Camera (NAC) imagery, the current resolution may be insufficient for applications requiring fine or spatially consistent discrimination, such as landing-site characterisation or sub-km surface feature analysis. Leveraging the patch-level embeddings from the same encoder is a natural approach to addressing higher-resolution downstream use cases; however, variability in spatial resolution across data modalities and latitudes poses additional challenges. More broadly, rigorous evaluation of representation quality in the absence of large labelled datasets is an open methodological challenge in planetary science. The PCA and UMAP analyses provide interpretable visualisations but do not constitute formal proof of geophysical fidelity, and establishing quantitative evaluation protocols is an important direction for follow-up work. Finally, the model architecture and hyperparameters were set based on prior work and practical constraints rather than domain-specific optimisation, and both have significant room for refinement in future iterations.

**Conclusion & Outlook.** LunarFM constitutes an initial effort toward a reproducible multimodal latent representation for the lunar ML community, integrating six physically distinct remote-sensing modalities into a shared representation space. The presented use cases demonstrate that these embeddings can be useful for low-label mineral mapping and similarity-based retrieval, and the open release of the dataset, model, embeddings, and notebooks is intended to lower the barrier for the broader community to explore and extend these capabilities, refine the framework, and identify the most promising directions for future work. Future work can build on this foundation in several directions. Extending geographic coverage to include polar regions would extend the framework to scientifically and operationally important areas of the Moon. Incorporating NAC-scale imagery, as well as other input datasets such as terrain elevation derivatives (slope, aspect, roughness) or hyperspectral imagery, and moving toward temporally-indexed inputs might better capture dynamics (e.g., thermal) and mission-dependent observation conditions. On the modelling side, alternative self-supervised objectives, such as contrastive learning or joint-embedding predictive architectures, may better disentangle correlated modalities such as the Diviner thermophysical channels, and could be explored in future iterations of the framework. Developing principled benchmarks for representation quality in low-label planetary science settings is a priority, including reference against unimodal and raw-feature approaches to quantify what multimodal pretraining specifically contributes. If the embeddings prove reliable for specific resource-oriented tasks, a natural next step is integrating them into broader analytical pipelines, coupling similarity search with expert-in-the-loop workflows for rapid site screening, or connecting the representation space to language-based retrieval for human-interpretable query interfaces. We view LunarFM not as a finished product but as an infrastructure layer, one that we hope future missions, datasets, and scientific questions can continue to build upon.

## 6 ACKNOWLEDGEMENTS

This work has been enabled by Frontier Development Lab Europe (https://fdleurope.org), a public/private partnership between the European Space Agency (ESA), the Luxembourg Space Agency, Datarock, Trillium Technologies, the University of Oxford and leaders in commercial AI supported by Google Cloud, SCAN and Nvidia, developing open science for all Humankind. Furthermore, the authors thank Valentin Bickel, David Briguglio, Katherine Hadler, Mike Heyns, Alex Maritati, Thomas Schaap, Ben Moseley, and James Parr for their guidance, discussions, and feedback throughout this research.

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

## A    DATASET SOURCES AND SPLITS

This appendix details the source products used to construct the LunarChips dataset and the additional datasets used in downstream experiments. All products were obtained from NASA's Planetary Data System (PDS) or from mission-associated lunar map repositories, and were subsequently reprojected, co-registered, resampled, and tiled into the common $0.5° \times 0.5°$ chip grid described in Section 3. The representation-learning stage uses the six primary sensing modalities described in Table 1; downstream labels and auxiliary layers are used only for evaluation and illustrative applications.

### A.1    PRIMARY INPUT MODALITIES USED FOR LUNARFM PRETRAINING

**LRO LROC WAC Hapke Photometric Maps (7 channels).**    The multispectral optical input is derived from the Lunar Reconnaissance Orbiter Camera (LROC) Wide Angle Camera (WAC) Hapke-normalised mosaic product, available through the LROC data portal. We use the seven-band UV–visible mosaic spanning 321–689 nm. This product provides photometrically normalised reflectance and is based on the Hapke parameter mapping framework described by Sato et al. (2017). In LunarFM, these bands form the main multispectral optical modality.

**LRO LOLA topography (1 channel).**    The topographic input is derived from the Lunar Orbiter Laser Altimeter (LOLA) gridded digital elevation model (DEM) distributed through the LRO/LOLA archive. LOLA provides the geodetic and topographic reference framework for the Moon. We use the global gridded elevation product as the single-channel topography modality.

**LRO Diviner thermophysical products (3 channels).**    The thermal modality includes Diviner Lunar Radiometer-derived products. In particular, we use global maps of regolith temperature, bolometric temperature, and rock abundance derived from long-term nighttime anisothermality observations. These products follow the updated global thermophysical mapping framework of Powell et al. (2023). In the model implementation, Diviner-derived inputs are grouped into thermophysical channels, with rock abundance treated separately in preprocessing/model grouping.

**LRO Mini-RF radar products (2 channels).**    The radar modality is derived from LRO Mini-RF products distributed through the PDS Geosciences Node. Mini-RF is a synthetic aperture radar instrument designed to characterise lunar surface roughness, blockiness, and radar backscattering properties (Nozette et al., 2010). The LunarFM input stack uses two radar-derived channels, namely the circular polarisation ratio (CPR) and the first Stokes parameter (S1), both resampled to the common chip grid.

**GRAIL gravity-derived products (4 channels).**    The gravity modality comprises products from the Gravity Recovery and Interior Laboratory (GRAIL) mission. These layers were assembled from GRAIL-derived gravity products, including free-air gravity anomaly, Bouguer anomaly, and additional gravity-related fields used as proxies for subsurface structure. The underlying gravity-field models are described by Goossens et al. (2020). In our data stack, four gravity-related channels are included: free-air anomaly, Bouguer anomaly, disturbance, and an uncertainty layer.

**Clementine UVVIS reflectance/albedo (1 channel).**    Clementine UVVIS products provide a multispectral global view of surface reflectance. For LunarFM, we include the 750 nm wavelength Clementine-derived reflectance channel, resampled to the common grid. The instrument and mapping quality are discussed in more recent reprocessing and geodetic reassessment work by Speyerer et al. (2023).

### A.2    ADDITIONAL DATASETS USED FOR DOWNSTREAM TASKS AND EVALUATION

**Lunar Prospector gamma-ray spectroscopy compositional maps.**    For mineral and elemental downstream tasks, LunarChips includes global compositional maps derived from Lunar Prospector Gamma-Ray Spectrometer (GRS) measurements. These maps provide estimates of major-element abundances and are based on the analysis framework of Prettyman et al. (2006). In our experiments, these layers are used only as supervised learning targets for downstream regression tasks.

**WAC-derived ilmenite map.** For downstream experiments related to ilmenite, we use the LROC WAC $TiO_2$ abundance product derived from UV/Vis reflectance. This map follows the methodology of Sato et al. (2017) and, in this work, is used as a label dataset for supervised learning $TiO_2$ mapping experiments.

**Crater annotations.** For crater-focused analyses, LunarChips incorporates crater annotations from the global lunar crater database of Robbins (2019) (craters $\geq$ 1–2 km in diameter). This database provides global crater locations and sizes and is also used for downstream supervised learning tasks.

**Expert-curated high-$TiO_2$ region delineations.** For the few-shot resource-screening experiment, we additionally use the high-$TiO_2$ region delineations proposed by Diaz & Keszthelyi (2025). In this work, these delineations are treated as an expert-curated reference set for simulating realistic low-label expert-guided sampling. These data are used exclusively in downstream experiments.

## A.3 DATASET SPLITS

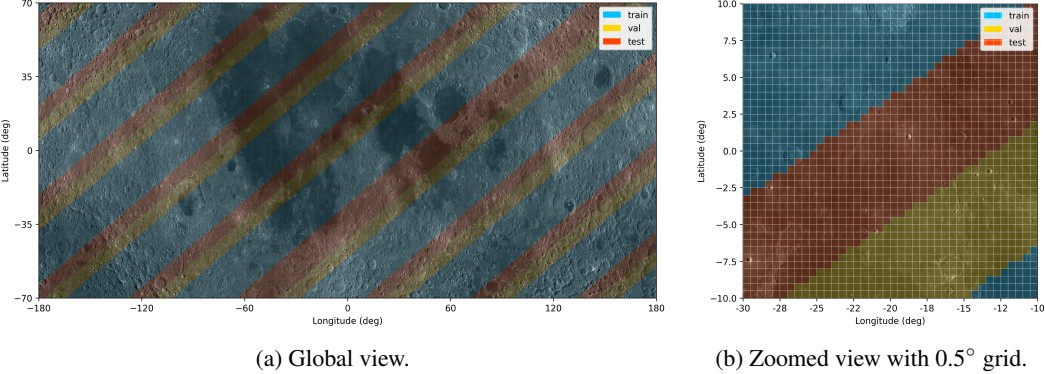

(a) Global view.

(b) Zoomed view with $0.5°$ grid.

Figure 10: Visualisation of the half-degree lunar grid and the diagonal band-wise train/validation/test split. **Left:** global WAC background with split overlay. **Right:** zoomed region showing the half-degree grid over WAC with the split overlay.

