# OpenReview forum: "LunarFM: a multimodal representation of the Moon's surface"
_ICLR.cc/2026/Workshop/FM4Science — ICLR 2026 Workshop FM4Science Poster_

### Official Review · Reviewer_ny9m · 2026-02-19
**Valuable Lunar Dataset and Domain-Appropriate FM Application**

**Rating:** 7
**Confidence:** 3

**Review:**

### Summary

This paper presents LunarFM, described as the first foundation model for multimodal lunar remote-sensing data. The system integrates six data modalities from three missions (LRO, GRAIL, Clementine) across 18 input channels, covering the Moon from -70° to +70° latitude. The architecture is a MultiMAE (Bachmann et al., 2022) — a multi-modal masked autoencoder with ViT backbone — trained with 85% masking on 0.5°×0.5° chips (201,600 total). The contribution is framed as a "data stack": (1) LunarChips, a co-registered multimodal dataset; (2) a pre-trained 109.8M-parameter model; (3) LunarEmbeddings, chip-level 768-dim representations; and (4) downstream applications including similarity search, unsupervised clustering, mineral regression (random forest on embeddings for FeO, TiO$_2$, MgO, etc.), and a few-shot TiO$_2$ mapping experiment using expert-curated samples from Diaz & Keszthelyi (2025). Evaluation covers reconstruction error (Figure 3), embedding analysis (PCA, UMAP, clustering), and downstream mineral prediction.

### Pros

**1. Genuinely novel application domain with strong workshop fit.** To my knowledge, no prior work has attempted a multi-task foundation model for lunar remote sensing using this breadth of instruments. Sander et al. (2025) used a multimodal transformer for lunar data, but focused narrowly on DEM/albedo reconstruction from a single task perspective, as the paper correctly notes. LunarFM's scope — six physically distinct instruments spanning optical, thermal, radar, topographic, gravitational, and reflectance modalities — is substantially broader and more aligned with the FM4Science workshop's theme of building domain-specific foundation models for science. This is exactly the kind of submission this workshop should attract.

**2. Thoughtful dataset curation.** The LunarChips dataset addresses a real pain point. Lunar data is genuinely fragmented across mission-specific archives with wildly different resolutions, formats, and coverage. Co-registering everything to a common 0.5° grid, handling the varying native resolutions, and providing a clean train/val/test split with diagonal banding to mitigate spatial leakage (Figure 2) reflects careful engineering. The inclusion of auxiliary evaluation layers (GRS-derived compositions, crater annotations, Diaz & Keszthelyi high-Ti delineations) that are kept separate from pretraining is methodologically sound.

**3. The embedding analysis is informative.** The progression from low embedding-dimension correlation ($|r| \geq 0.5$ for only 2.8% of pairs), through PCA (54.1% variance in 4 components with geologically interpretable spatial patterns), UMAP (continuous manifold with TiO$_2$-separated subregion), to k-means clustering (spatially contiguous clusters without neighbourhood constraints) builds a coherent story that the representation captures meaningful lunar surface variability. The fact that cluster boundaries align with mare/highland boundaries despite no coordinate input is a reasonable sanity check.

**4. The few-shot TiO$_2$ experiment is well-designed.** Using only 5 expert-selected positive chips (from high-Ti regions per Diaz & Keszthelyi 2025) and 5 negative chips, repeated 20 times, with comparison to random selection (correlation $0.784 \pm 0.025$ vs. $0.653 \pm 0.267$) is a realistic scenario for planetary science and demonstrates practical utility. The 10× reduction in variance with expert selection is a nice finding.

**5. Open-source commitment.** The promise to release the dataset, model, embeddings, and tutorials is commendable and would genuinely benefit the (small) community working at the intersection of ML and lunar science.

**6. Honest about limitations.** The Discussion acknowledges that it remains unclear whether the representation isolates specific resource signatures vs. broader terrain patterns, and flags robustness to missing modalities and domain shift as open questions. This self-awareness is appreciated.

### Cons

**1. Zero baselines.** This is the most significant weakness. The paper reports no comparison to any alternative approach. Natural baselines include: (a) a single-modality MAE (e.g., trained only on WAC or only on LOLA) to quantify the benefit of multimodal pretraining; (b) the raw 18-channel input directly into a random forest (to quantify the benefit of representation learning over raw features); (c) a simple PCA baseline on the concatenated raw channels; (d) an ImageNet-pretrained ViT applied to the optical channels; or (e) an Earth observation FM such as DOFA (Xiong et al., 2025) or a pretrained TerraMind adapted to the optical/DEM bands. Without any of these, the reader cannot assess whether the learned embeddings provide meaningful compression beyond what trivial approaches would achieve. For example, the mineral regression MAE values in Figure 9 are reported but never contextualized — are they good? Better than raw features? Better than a simple spectral index? The few-shot TiO$_2$ correlation of 0.784 is encouraging, but what would a spectral ratio model trained on the same 10 chips achieve?

**2. No architectural novelty.** The model is a direct application of MultiMAE (Bachmann et al., 2022) with modality-specific linear projections. The only design decisions are the patch size (8×8), chip resolution (112×112), masking ratio (85%), and the grouping of Diviner into two modalities. These are hyperparameter choices, not architectural contributions. The paper does not justify why MultiMAE is preferred over alternatives such as CROMA's contrastive objective, OmniSat/AnySat's JEPA-based approach, or TerraMind's dual-scale token+pixel fusion. A brief ablation or comparison among pre-training strategies would substantially strengthen the paper.

**3. Evaluation is largely qualitative.** Most of the evaluation relies on visual inspection: reconstruction examples (Figure 4), PCA maps (Figure 5), UMAP plots (Figure 6), cluster maps (Figure 7), and similarity search results (Figure 8). These are useful for building intuition but are not rigorous evaluations. The quantitative metrics that do appear — reconstruction MSE (Figure 3), mineral MAE (Figure 9), and few-shot correlation (Figure 10) — are never compared against any baseline or reference value. The reconstruction MSEs (e.g., 0.092 for Diviner RA, 1.275 for MiniRF) lack units or normalization context — is 0.092 relative to what dynamic range? How does this compare to reconstructing from the training-set mean?

**4. Downstream tasks are shallow.** All downstream experiments use frozen embeddings with a random forest. There is no fine-tuning evaluation — no linear probing, no end-to-end fine-tuning with a task head, no comparison of frozen vs. fine-tuned representations. The GFM literature (PANGAEA benchmark, GEO-Bench) has established standard protocols for evaluating foundation model representations that involve at least linear probing; random forest on a global embedding is the weakest possible test. Additionally, the mineral regression uses 20,000 training chips (~10% of data), which is generous — most FM papers evaluate at 1%, 5%, and 10% to show data efficiency curves.

**5. MiniRF handling raises concerns.** Figure 1(d) shows MiniRF has substantial spatial gaps (NaN values), and the reconstruction MSE is an order of magnitude worse than other modalities (1.275 vs. 0.001–0.092 with modality present; 1.262 vs. 0.007–0.266 without). The paper acknowledges this but does not address how NaN values are handled during training — are they masked, zero-filled, or interpolated? This matters because if NaN regions are masked out, the model may simply learn to predict zeros for MiniRF, inflating cross-modality reconstruction metrics for other modalities. An ablation training with and without MiniRF would clarify its contribution.

**6. Coarse spatial resolution may limit practical utility.** Each chip covers 0.5° of latitude and longitude, which corresponds to roughly 15 km at the equator. The chip-level embedding collapses all spatial information within this footprint into a single 768-dim vector. For resource mapping at the resolution relevant to landing site selection or ISRU operations (meters to hundreds of meters), this is extremely coarse. The paper does not discuss whether patch-level embeddings (which preserve 14×14 spatial resolution within each chip) could be used for finer-grained tasks, nor whether the approach could be applied at higher resolution with smaller chips.

**7. Missing ablation studies.** No experiments vary the masking ratio (85% is borrowed from MultiMAE without justification for the lunar domain), the number of input modalities (which modalities contribute most?), the model size (109.8M parameters — is this overparameterized for the data size?), or the chip resolution. These are standard in FM papers and their absence makes it difficult to understand which design decisions matter.

**8. Anonymity concerns.** The FDL LunarLab project has publicly announced "Lunar-FM" as "the first AI foundation model of the Moon" (Luxembourg Space Agency press release), describing a system with the same specifications: 18 data layers from LRO/GRAIL/Clementine, 768-dimensional embeddings, ±70° latitude coverage, Vision Transformer architecture, TiO$_2$ few-shot mapping with expert-curated samples, and similarity search capabilities. The overlap in technical specifications between this submission and the publicly described Lunar-FM project may compromise the double-blind review process.

**9. Data and model not yet available.** The abstract states resources "will be made publicly available in the near future." For a paper whose primary contribution is framed as a data stack and open-source framework, the inability to verify any artifacts at review time is a meaningful limitation. The GFM community has increasingly expected code/data availability at submission (TerraMind, DOFA, AnySat all released code and weights with their papers).

---

### Official Review · Reviewer_CxPx · 2026-02-24
**Multimodal Lunar Representation Learning: Strong Illustrative Results with Limited Quantitative Evaluation**

**Rating:** 6
**Confidence:** 4

**Review:**

The manuscript presents a novel foundation model for multimodal lunar surface representation learning using a multimodal masked autoencoder. Several downstream applications are explored. The work is illustrative and contains many interesting examples; however, certain aspects would benefit from additional clarification and quantitative evaluation to strengthen the claims.

Major Comments

1.	It appears that all modalities are represented as spatial tensors (e.g., 112×112 with varying channel counts). It would be helpful to explicitly describe the tensor structure and value ranges for each modality, as this would clarify preprocessing choices and reconstruction difficulty.

2.	The dataset is restricted to −70° to +70° latitude, presumably to ensure overlapping coverage across modalities. Could the authors clarify how much spatial coverage was excluded due to requiring full modality intersection? Additionally, would pretraining on subsets of modalities allow larger spatial coverage? A modality ablation study with maximal available coverage per subset could provide insight into the trade-off between multimodal integration and data availability.

3.	The reconstruction error for the MiniRF modality is not only significantly higher than for other modalities (as noted by the authors), but also appears nearly unchanged when the modality is fully masked. This raises the question of whether MiniRF provides meaningful signal to the shared representation, or whether noise, preprocessing artifacts, or weak cross-modal predictability limit its contribution. It would be informative to comment on this behavior and possibly evaluate performance without this modality.

4. In contrast, the LRO DEM modality exhibits low reconstruction error and limited sensitivity to masking. Could the authors comment on whether this reflects strong cross-modal predictability, intrinsic spatial smoothness, or properties of the MSE objective?

5. In Section 4.2, the principal components are interpreted as corresponding to large-scale lunar surface features. However, this interpretation appears qualitative. It would strengthen the claim to provide quantitative evidence (e.g., correlation with elevation, gravity anomalies, or known geological maps), and to clarify the interpretation of PC2 in particular, as it seems different to the remaining 3 PCs, albeit accounting for 14% of the variance.

6. The UMAP visualizations are described as aligning with surface features and geological regions. Since UMAP is primarily a visualization tool, it would be beneficial to quantify this alignment as in the previous point.

7. The similarity search examples are visually compelling but would benefit from quantitative evaluation. Additionally, retrieval is demonstrated for four modalities; does performance differ for the remaining modalities, and if so, why?

8.	The first downstream task appears to be evaluated using a single fine-tuning run. While random forests are generally stable on large datasets, reporting results across multiple random seeds would help quantify the variance introduced by bootstrap sampling and feature randomness.

9.	It would be valuable to analyze modality contributions to downstream tasks, particularly mineral prediction. An ablation study could clarify which modalities drive performance gains.

---

### Official Review · Reviewer_3vDZ · 2026-02-25
**Review of LunarFM: a multimodal representation of the Moon's surface**

**Rating:** 6
**Confidence:** 4

**Review:**

The paper proposes LunarFM, a foundational model for the moon, that aims to combine data from multiple lunar instruments, combining six data modalities from three missions (18 channels). They introduce an ML-ready dataset  (LunarChips), a ready-to-use ML embedding dataset (LunarEmbeddings), and ready to use version of the pre-trained model. Prior FM work includes Geospatial Foundation Models (GFM), used for earth observation; this is the first attempt to do so with lunar data and aims to scale multimodal lunar analysis and enable resource assessment workflows.

**Strengths:**
* The application domain is novel, and combining and harmonizing data from multiple missions into a unified dataset is valuable
* Love that the work makes available a dataset, an embedding dataset, and ready to use version of the model, along with tutorials to enable building upon and advancing this work
* Qualitative reconstructions show the model learns both intra-modality and cross-modality correlations, showing the representative power of the embeddings
* The PCA and UMAP analysis is well motivated, showing the representative power of the embeddings
* The example downstream use case (e.g., TiO2) helps show the utility and application of the embeddings

**Weaknesses:**
* There is no real benchmarking of other embedding approaches or comparing it with using an Earth Embedding model and applying it to the Moon, would be good to include this to see that the architectural choice performs well on downstream tasks. Hard to appreciate the reconstruction numbers without any comparison.
* There could be more ablations carried out to justify model choices e.g. an ablation contrasting global token vs pooled patch embeddings
* While two tasks are shown, claiming that this is a FM would require several more tasks to show the generalizability of the model, and would need some comparison with approaches that might be optimized for just 1 task.
* Section 6 on Outlook acknowledges this, but it’s hard to consider this an FM without any temporal indexing.

**Clarification:**
* How were modalities normalized and how was the reconstruction loss weighted across modalities to make MSEs comparable and avoid domination by high-variance channels?
* The MiniRF reconstruction error is roughly an order of magnitude higher than other modalities both with and without the modality present at input. Does the model effectively leverage MiniRF information during encoding, or does the high error suggest it is largely ignored? Have you analyzed whether removing MiniRF from the input changes downstream task performance?

**Minor:**
* There are some spelling/grammar errors on Line 450: “LunarFM is accompanied [insert by] LunarChips, a multimodal dataset of half degree chips covering the lunar surface between -70° and +70° latitude across siz[should be six] instruments (18 channels)” - please fix for the Camera Ready version.

**Overall Decision:**  6/10 - weak accept. The application domain is interesting and timely, with renewed interest in the Moon. The work would be strengthened with more comparisons and evaluations with other approaches e.g. other embedding architectures and also evaluation on more downstream tasks to strengthen the claim that it is an FM. The dataset and approach to building an FM for the moon are compelling, so it merits a weak accept for a workshop paper, especially with the sharing of a dataset, embeddings, and a model.

---

### Official Review · Reviewer_XymE · 2026-02-25
**This paper introduces LunarFM, a multimodal foundation model for lunar remote sensing that integrates heterogeneous data from six instruments across three lunar missions into a shared embedding space. The authors propose a self-supervised MultiMAE-based architecture trained on co-registered 0.5° × 0.5° lunar “chips” and release both the pretrained model and a companion embedding dataset. The work aims to support a range of downstream lunar science tasks, including similarity search and resource mapping (e.g., TiO₂).**

**Rating:** 5
**Confidence:** 3

**Review:**

Summary:
The idea of a lunar-scale foundation model is novel and impactful. However, while the conceptual contribution and dataset construction are compelling, the empirical evaluation is currently insufficient to substantiate several of the paper’s claims, and key details regarding data provenance and preprocessing are missing. As a result, I find the paper promising but not yet convincing for acceptance in its present form.

Strengths
- Novel problem setting: To my knowledge, this is one of the first attempt to develop a general-purpose multimodal foundation model for lunar science, which is well aligned with renewed interest in lunar exploration and resource assessment.

- Ambitious multimodal integration: The paper integrates optical, topographic, thermal, radar, and gravity data across multiple missions, which is technically challenging and valuable.

- SSL approach: The use of a MultiMAE-style masked autoencoder is appropriate for low-label planetary science data.

- Potentially valuable data products: The proposed LunarChips and LunarEmbeddings datasets, if released with sufficient documentation, could be useful resources for the community.


Weaknesses

- Insufficient description of data sources and preprocessing
- Too much focus on Introduction & Related work for such a short paper
- The paper does not clearly specify where the training data is sourced from (e.g., exact repositories, processing levels, versions, etc). This is a major concern given that dataset construction is a core contribution.
- Lacking technical depth: Preprocessing steps are not described in sufficient detail to ensure reproducibility.
- Over-interpretation of embedding correlation analysis: The claim that low pairwise Pearson correlation between embedding dimensions indicates “maximised information” is not justified.
- Low linear correlation only demonstrates lack of trivial linear redundancy and does not imply information richness necessarilly
- This analysis is closer to a sanity check than a substantive evaluation of representation quality.
- Weak Downstream Evaluation: Mineral mapping experiments rely on a single downstream model (random forest) with limited justification.
- It is unclear whether improvements arise from multimodal fusion, spatial autocorrelation, or coarse geological proxies.
- Heavy dependence on expert selection in few-shot experiments: Performance improves substantially only when expert-curated samples are used.

This weakens claims about general-purpose few-shot capability.
- Missing ablations and robustness analyses
- No analysis of modality importance, robustness to missing modalities at inference, or sensitivity to embedding dimensionality.

Questions for the Authors

- Can the authors provide a precise list of data products, versions, and repositories used for each modality, along with preprocessing details?
- How do the downstream results compare to simpler baselines such as PCA on concatenated inputs or unimodal models?
- Have the authors evaluated effective dimensionality (e.g., eigenvalue spectra) or conducted ablation studies on the embedding size?

Overall Assessment

The paper presents a strong and interesting idea with clear potential impact, but the current version lacks the technical depth and transparency needed to support its claims. In particular, clearer data provenance, stronger baselines, and more principled evaluation of the learned representations are necessary.

---

### Meta-Review · Area_Chair_kwkv · 2026-02-27

**Recommendation:** Accept (Poster)
**Confidence:** 4

**Metareview:**

This submission has received four reviews. Three positive reviews with a "accept" and two "marginally above acceptance threshold". One reviewer rated the paper with a "marginally below acceptance threshold".

After reading the reviews, I recommend this paper for "acceptance" and ask the authors to consider implementing the feedback given by all reviewers (especially reviewer XymE) into the camera-ready version of the paper.

---

### Decision · Program_Chairs · 2026-03-03

Accept (Poster)